# Analysis of the Incidence of Type 2 Diabetes, Requirement of Insulin Treatment, and Diabetes-Related Complications among Patients with Cancer

**DOI:** 10.3390/cancers15041094

**Published:** 2023-02-08

**Authors:** Su Jung Lee, Chulho Kim, Hyunjae Yu, Dong-Kyu Kim

**Affiliations:** 1School of Nursing, Research Institute of Nursing Science, Hallym University, Chuncheon 24252, Republic of Korea; 2Department of Neurology, Chuncheon Sacred Heart Hospital, Hallym University College of Medicine, Chuncheon 24252, Republic of Korea; 3Institute of New Frontier Research, Division of Big Data and Artificial Intelligence, Hallym University College of Medicine, Chuncheon 24252, Republic of Korea; 4Department of Otorhinolaryngology-Head and Neck Surgery, Chuncheon Sacred Heart Hospital, Hallym University College of Medicine, Chuncheon 24252, Republic of Korea

**Keywords:** insulin, diabetes mellitus, type 2, cohort studies, early detection of cancer, neoplasms

## Abstract

**Simple Summary:**

Previous epidemiological studies have shown that type 2 diabetes may be associated with an increased risk of several cancer types and cancer-related mortality. However, whether cancer causes an increased risk of developing type 2 diabetes or its related complications remains unclear. The objective of this research was to investigate the risk of type 2 diabetes development, insulin requirements, and diabetes-related complications in patients with cancer. We found that cancer was associated with an increased risk of developing type 2 diabetes. Moreover, the number of new cases that required insulin was significantly higher in patients with cancer than in those without. Although there was no significant difference in the incidence of diabetes-associated complications between the two groups, some cancer types were associated with an increased risk of developing diabetic nephropathy. The findings suggest that clinicians should monitor the development of type 2 diabetes and its related morbidities in patients with cancer.

**Abstract:**

This retrospective nationwide population-based cohort study used a dataset collected from the Korean National Health Insurance Service. We evaluated incident type 2 diabetes, insulin requirements, and diabetes-associated complications during a 10-year follow-up period using the log-rank test and Cox proportional hazards regression models. In total, 8114 and 16,228 individuals with and without cancer, respectively, were enrolled. We found a higher incidence rate and an increased adjusted hazard ratio (HR) for new cases of type 2 diabetes in patients with cancer, compared with those without cancer. Additionally, patients with cancer had a higher risk of insulin requirement than patients without cancer (adjusted HR 1.43, 95% confidence interval [CI], 1.14–1.78). Although there was no significant association between diabetes-associated complications and overall cancer diagnosis, specific cancer types (pancreas, bladder, and prostate) showed an increased risk of subsequent diabetic nephropathy. Therefore, clinicians should closely monitor patients with cancer for the early detection of type 2 diabetes and related morbidities.

## 1. Introduction

Cancer is a global health problem, and it is estimated that approximately 19.3 million new cancer cases have been diagnosed [1]. With advances in cancer diagnosis and treatment techniques, people are able to live longer after a cancer diagnosis [2]. This prolongation of survival period has raised interest in the long-term quality of life and exacerbation of existing chronic diseases or the morbidity of new chronic diseases due to various treatments [3]. Among cancer-related morbidities, type 2 diabetes shares many risk factors with cancer, but the potential biological link between the two diseases is not fully understood [4]. In some cancers, poor blood sugar control increases the risk of cancer recurrence [5] and negatively affects quality of life [6]. This suggests that early detection, treatment, and management of type 2 diabetes are very important in cancer survivors. Current consensus recommendations for the management of hyperglycemia in patients with type 2 diabetes suggest lifestyle changes and monotherapy, preferably with metformin [7]. However, if the glycated hemoglobin (HbA1c) target is still not achieved, combination injectable therapy including basal insulin can be considered to obtain glycemic control.

Although some studies have reported that specific cancers, such as breast, colorectal, and pancreas, are related to the risk of developing diabetes [8,9,10], some cross-sectional studies [11,12] and cohort studies [13,14,15] described that patients diagnosed with cancer had a clear increase in the subsequent risk of type 2 diabetes. Besides the association between the incidences of two diseases, in the real world it is often observed that diabetes patients diagnosed with cancer tend to neglect the self-monitoring of blood glucose and do not adhere to diabetes medications [16,17,18,19]. Previously, a systematic study reported a general decline in diabetes management after a cancer diagnosis because the primary concern shifted from diabetes to cancer [20]. However, only a few studies have been conducted on the risk of requiring insulin treatment and complications related to newly developed type 2 diabetes among patients with cancer based on a nationwide population [14].

Therefore, in this study, we investigated whether the risk of diabetes increased after the onset of cancer using a propensity-matching analysis method that controlled for several confounding variables. Additionally, we assessed the risk of insulin requirement and diabetes-associated complications in cancer survivors, compared with participants without cancer during a 10-year follow-up period.

## 2. Materials and Methods

This cohort study used a nationwide population-based cohort dataset constructed from national health claims data collected by the Korean National Health Insurance Service (KNHIS). The Institutional Review Board (IRB) of Hallym Medical University, Chuncheon Sacred Hospital (Chuncheon, Republic of Korea), approved this retrospective, nationwide propensity score-matched cohort study (IRB No. 2021-08-006). The IRB waived the need for written informed consent because this study obtained secondary data from the KNHIS database after de-identification. The original datasets are not publicly available because of KNHIS policies; however, if the corresponding author deems it reasonable, the approach of generating and/or analyzing data in the present study could be made available.

### 2.1. Study Sample

We derived the study population from participants in a nationwide representative cohort sample of 1,025,340 adults from the KNHIS healthcare claims dataset from 2002 to 2013 [21,22]. South Korea has a single-payer national health system (KNHIS), covering the entire South Korean population since 1989. An insured individual pays for national health insurance, which is proportional to the individual’s income. Although a user charge exists, it is mandatory for all Koreans to join the KNHIS. Thus, the KNHIS includes major healthcare information such as inpatient and outpatient visits, procedures, and prescriptions. Additionally, a unique identification number is assigned to each South Korean at birth; hence, the claims data in the KNHIS cannot be omitted or duplicated. Therefore, the KNHIS cohort sample could reflect the entire Korean population and minimize selection bias.

### 2.2. Study Design

A schematic of the study design and enrollment of the study participants is shown in Figure 1.

Briefly, we had a washout period of one year (January to December 2002) to exclude those with a risk of developing type 2 diabetes before cancer diagnosis. We then identified patients with cancer using the diagnostic codes during the index period (between January 2003 and December 2005). We defined the cancer group using the presence of the same C code more than three times within a year or inpatient hospitalization with a C code. During this process, we carefully reviewed the C code because sometimes it is entered as a diagnosis code before biopsy results for insurance coverage, which can lead to confusion in determining the diagnosis date. Finally, we excluded patients (1) aged <15 years, (2) who died during the index period, and (3) who were diagnosed with type 2 diabetes before the diagnosis of cancer. Next, to select the comparison group (non-cancer), we randomly selected propensity score-matched participants from the remaining cohort registered in the database as two participants without cancer for each patient with cancer. Thus, each participant in the comparison group was matched with each patient in the cancer group based on all independent variables and year of enrollment (cancer diagnosis).

In this study, we defined the primary endpoints as incident development of specific events, such as type 2 diabetes, initial insulin treatment, and diabetes-associated complications. Additionally, we performed follow until the secondary endpoint, all-cause mortality, was reached. For patients who had no specific events (primary endpoints and mortality) and were alive on 31 December 2013 (the final follow-up day for this database), the data were censored at this time point (Appendix A). The database consisted of claim data during 2002–2013; thus, this does not include the data for the novel non-insulin antidiabetic drugs because this database only includes before the approval of these.

### 2.3. Independent Variables

We obtained detailed characteristics of each patient, including age, sex, residence, household income, and comorbidities. Specifically, we adjusted for comorbidities using the Charlson comorbidity index (CCI), which is a weighted index for categorizing comorbidities of patients. Thus, we classified all independent variables into three categories: age (15–39, 40–64, >64 years), residence (1st areas: Seoul, the largest metropolitan region in South Korea; 2nd area: other metropolitan cities in South Korea; and 3rd area: small cities and rural areas), household income (low: ≤30%, middle: 30.1–69.9%, and high: ≥70% of the median), and three comorbidity statuses (CCI: 0, 1, and ≥2).

### 2.4. Statistical Analysis

We assessed the incidence rate as a measure of the frequency at which type 2 diabetes or other incident events (initial insulin treatment and diagnosis of diabetes-associated complications) appeared over the follow-up period. The overall incidence rate was expressed as per 1000 person years: first, if the participant died, the number of years from the initial cancer diagnosis to the date of death; second, if specific events appeared, the number of years from the initial cancer diagnosis to the date of the first diagnosis of specific events; and finally, if there are no events, the number of years from the date of initial cancer diagnosis to the final follow-up day. In addition, to identify whether cancer could increase the risk of the occurrence of specific diseases, we used Cox proportional hazard regression analyses to calculate hazard ratio (HR) and 95% confidence interval (CI), adjusted for the other independent variables. The R software program (version 4.0.0; R Foundation for Statistical Computing, Vienna, Austria) was used for all statistical analyses.

## 3. Results

### 3.1. Cohort Sample Characteristics and Incidence of Diabetes

In the present study, we selected a cohort consisting of 8114 and 16,228 patients with and without cancer, respectively. We observed no significant differences in all independent variables between the two groups, indicating that each variable was appropriately matched (Appendix A). Regarding the analysis of incidence, we examined a total of 46,032.1 person years and 135,352.5 person years in the cancer and non-cancer groups, respectively. The overall incidence of type 2 diabetes was 21.18 per 1000 person years in the cancer group and 17.13 per 1000 person years in the non-cancer group.

### 3.2. Risk of Diabetes Development in Patients with Cancer

We analyzed the HR for incident type 2 diabetes events using univariate and multivariate Cox regression models. The cancer group showed a significant association with the subsequent development of type 2 diabetes (adjusted HR = 1.29, 95% CI = 1.20–1.39) (Figure 2).

Additionally, when the risk of type 2 diabetes was evaluated according to the type of cancer, the association was particularly strong for pancreatic cancer (adjusted HR, 2.33; 95% CI, 1.42–3.81). The adjusted HR for incident type 2 diabetes events was also significantly increased for esophagogastric cancer (adjusted HR, 1.37; 95% CI, 1.19–1.579), colorectal (adjusted HR, 1.29; 95% CI, 1.08–1.53), gallbladder cancer (adjusted HR, 1.80; 95% CI, 1.17–2.7), lip/oral/pharynx cancer (adjusted HR, 1.36; 95% CI, 1.12–1.67), bladder cancer (adjusted HR, 1.69; 95% CI, 1.23–2.32), prostate cancer (adjusted HR, 1.60; 95% CI, 1.21–2.12), and other unspecified cancers (adjusted HR, 1.52; 95% CI, 1.16–1.99). Therefore, to confirm the risk of type 2 diabetes diagnosis in patients with cancer, we performed a sensitivity analysis using a cohort dataset excluding pancreatic cancer (Table 1).

The adjusted HR for type 2 diabetes development was significantly higher in the cancer group (without pancreatic cancer) than in the non-cancer group (adjusted HR, 1.28; 95% CI, 1.19–1.38). Specifically, we found that male and middle-aged or older patients had a significantly higher likelihood of developing type 2 diabetes (female: adjusted HR = 1.22 [95% CI, 1.09–1.36], age 40–64 years: adjusted HR = 1.43 [95% CI, 1.29–1.58], and age >64 years: adjusted HR = 1.14 [95% CI 1.01–1.28]) (Appendix A).

### 3.3. Risk of Diabetes Treated with Insulin Therapy in Patients with Cancer

To identify the risk of insulin requirement, we selected patients with diabetes who were receiving insulin treatment. The incidence rate of newly developed cases requiring insulin treatment was 2.34 per 1000 person years in the cancer group and 1.74 per 1000 person years in the comparison group. The adjusted HR for insulin requirement among patients with diabetes associated with cancer development was 1.43 (95% CI, 1.14–1.78) (Figure 3).

Moreover, when the risk of insulin requirement was evaluated by type of cancer, we detected a significantly increased adjusted HR for liver, gallbladder, pancreas, lung, and hematologic cancers. Among these, pancreatic cancer showed the highest risk of insulin requirement. Specifically, the risk of requiring insulin treatment after gallbladder and pancreatic cancer development was higher in the first year after cancer diagnosis, whereas hematologic cancer showed a relatively lower risk of requiring insulin treatment after cancer development during the three years (Figure 4).

To confirm these findings, we performed a sensitivity analysis using a cohort dataset for all cancer types, except pancreatic cancer (Table 2).

In this sensitivity analysis, we detected an increased adjusted HR for type 2 diabetes requiring insulin treatment in the cancer group (except pancreatic cancer), compared with the non-cancer group (adjusted HR, 1.35; 95% CI, 1.07–1.69).

### 3.4. Risk of Diabetes-Associated Complications in Patients with Cancer

To evaluate the effect of cancer on diabetic comorbidities, we defined diabetes-associated complications as the sum of events of diabetic nephropathy, diabetic retinopathy, diabetic neuropathy, and diabetic arthropathy. We assessed the incidence rate and risk of diabetes-associated complications according to cancer type (Figure 5).

We could not find an association between overall cancer diagnosis and diabetes-associated complications; however, some cancer types (pancreas, breast, bladder, and prostate) showed a significant adjusted HR for the development of diabetes-associated complications. Among those, the risk of diabetes-associated complications increased in the three years after pancreatic cancer diagnosis and in the five years after prostate cancer diagnosis. The risk ratios of diabetes-associated complications for breast and bladder cancers were relatively consistent throughout the follow-up period (Figure 6).

The subgroup analysis of diabetes-associated complications showed no significant association between overall cancer and diabetic nephropathy, diabetic retinopathy, diabetic neuropathy, or diabetic arthropathy (Appendix A). However, we detected that certain cancers (pancreas, breast, bladder, and prostate) showed a significant adjusted HR of 4.79 for diabetic nephropathy, but there was no significant difference in diabetic retinopathy, diabetic neuropathy, and diabetic arthropathy, respectively, between the groups (Figure 7).

## 4. Discussion

In this longitudinal cohort study, we found that patients diagnosed with cancer had a significantly higher risk of incident type 2 diabetes than those without cancer. Interestingly, we also found that the incidence rate and adjusted HR for newly developed diabetes cases requiring insulin treatment in the cancer group were significantly higher than those in the non-cancer group. Additionally, some cancer types, including pancreas, bladder, and prostate, led to an increased risk of subsequent diabetic nephropathy development during the follow-up period, although there was no association between overall cancer development and diabetes-associated complications.

Currently, the underlying mechanism for the significantly increased risk of developing diabetes in cancer survivors remains unclear [23]. However, various cancer treatment methods, such as radiation therapy, chemotherapy, surgery, hormone therapy, and the use of corticosteroids, may be associated with the development of diabetes [24,25,26,27]. It is known that type 2 diabetes highly influences the health status of patients with cancer. A study reported that poor blood sugar control could increase the risk of bladder cancer recurrence [5]. Another study showed that breast cancer survivors experienced poor quality of life because of uncontrolled glucose levels [6]. Moreover, poorly controlled diabetes is a major risk factor for cardiovascular disease and a major non-cancer-related cause of death in cancer survivors [28,29,30]. Our findings are consistent with those of previous studies that reported a higher risk of developing type 2 diabetes in cancer survivors than in the general population, based on population-based cohort studies and a systematic literature review with meta-analysis [23,31,32].

Besides the increase in the subsequent risk of diabetes, we found a clear relationship between increased requirement of insulin treatment in cancer survivors. A previous study showed that hyperglycemia occurs in approximately 10–30% of patients during chemotherapy [33]. In particular, L-asparaginase, a cytotoxic chemotherapeutic agent, has a direct toxic effect on pancreatic beta cells through the inhibition of insulin production and release, and an indirect contribution to hyperglycemia is associated with the development of pancreatitis. In addition, the function of such damaged beta cells can continue even after chemotherapy is terminated [34].

Several other studies have demonstrated that everolimus and temsirolimus are highly associated with hyperglycemia and poor glucose control in patients with cancer [33,35]. Androgenic monotherapy, a common treatment for prostate cancer, is also associated with the occurrence of type 2 diabetes, and the duration of androgenic monotherapy significantly affects the risk of developing poorly controlled diabetes, peaking at three years of treatment [36,37]. Thus, hyperglycemia or poor blood sugar control for cancer-specific treatment may contribute to the need for insulin treatment in cancer survivors. Interestingly, we also found a clear increase in the risk of requiring insulin treatment in cancer survivors, except for patients with pancreatic cancer. This means that patients with cancer may be vulnerable to hyperglycemia and poor control of glucose levels.

In general, microvascular complications in type 2 diabetes include nephropathy, neuropathy, and retinopathy. These complications are related to age, duration of diabetes, body mass index, high blood pressure, and dyslipidemia [38,39]. Previously, a study demonstrated an increased risk of preventable diabetic complications (diabetic emergency visits, skin and soft tissue infections, and cardiovascular disease) in patients with breast, prostate, and colorectal cancers compared with controls [40]. Another study revealed that long-term micro- and macrovascular complication risk regarding preexisting diabetes after cancer was similar between patients with and without cancer [41]. Therefore, there have been conflicting reports on whether cancer survivors with diabetes are more susceptible to the microvascular complications of type 2 diabetes. Consistent with these findings, our findings showed no significant association between cancer diagnosis and increased risk of diabetes-associated complications. However, one hypothesis that may explain the association between a higher risk of diabetic nephropathy in cancer survivors is that cancer survivors are generally more likely to have cancer-related diseases, such as cachexia, hyperuricemia, and hypercalcemia. Thus, they appear to be more susceptible to renal dysfunction and damage [42].

Moreover, cancer survivors are more likely to have received at least one cancer treatment, which may cause a glomerular filtration rate reduction, albuminuria, nephrotoxicity, and proteinuria. Several studies have revealed that some cancer survivors have residual kidney damage, which may contribute to an increased risk of diabetic nephropathy in cancer survivors [43,44]. Similarly, we found that some cancer types were associated with an increased risk of diabetic nephropathy development during the follow-up period.

The results of this study require careful interpretation, considering the following limitations: First, our database was that of a cohort sample, and only a limited number of identifiable variables were included. Therefore, we could not correct for lifestyle risk factors that could affect the occurrence of diabetes, such as body mass index, smoking history, and family history of diabetes. Second, we depended on diagnostic codes to classify type 2 diabetes and identify diabetes-related complications. Therefore, it is possible that patients with undiagnosed type 2 diabetes or those with less severe diabetic complications, limited access to medical centers, or who did not seek medical advice were excluded from this study. Third, the two groups were compared by determining the severity of diabetes only based on whether insulin was used. According to the recommendations of the American Diabetes Association, insulin treatment should be given if HbA1c does not reach the appropriate target despite the use of two or more oral hypoglycemic agents. However, in this study, we could not directly determine the degree of blood sugar control level because of the absence of HbA1c levels in our database. In addition, treatment-related variables, such as chemotherapy and radiation therapy, which can play an important role in the development of diabetes after cancer diagnosis, cannot be confirmed using our database. It should be noted that there may be unmeasured confounders related to the treatment. Finally, this database does not include cancer characteristics such as histologic type, stage at diagnosis, or classification according to the TNM classification system.

Nevertheless, our study has several advantages. This was a population-based study, and patients who developed diabetes before cancer were excluded after a sufficient washout period. In addition, we matched several important variables that could affect the incidence of diabetes in both groups using propensity scores. Thus, we could minimize the effect of lifestyle and behavior differences between different age groups, sexes, income groups, and regions. Additionally, we wanted to eliminate the effect of diseases other than cancer on the analysis of the risk of incident development for type 2 diabetes, requirement of insulin, and diabetes-associated complication events. Therefore, in this study, we matched the overall comorbidity status between the two groups using the CCI. The CCI, the most ubiquitous comorbidity risk score, predicts one-year mortality among hospitalized patients and provides a single aggregate measure of patient comorbidity. Several recent claim data studies have also used the CCI to match the comorbidities status between the control and experimental groups [45,46,47,48,49]. Finally, the participants of this study were a single ethnic cohort, and the effect of important variables on the incidence of diabetes, such as race, was minimized.

## 5. Conclusions

This study investigated the possible link between cancer and the prospective development of type 2 diabetes, insulin treatment, and diabetes-associated complications during a 10-year follow-up period. Interestingly, we observed that patients with cancer had a significantly higher risk of developing type 2 diabetes and insulin treatment than those without during a 10-year follow-up period. Additionally, some cancer types are significantly associated with an increased risk of diabetic nephropathy. Therefore, we suggest that clinicians should consider that patients with cancer may have a higher risk of developing type 2 diabetes and require insulin treatment. Additionally, clinicians should be aware of the potential risk of developing diabetic nephropathy in patients who suffer from pancreatic, bladder, and prostate cancer.

## Figures and Tables

**Figure 1 cancers-15-01094-f001:**
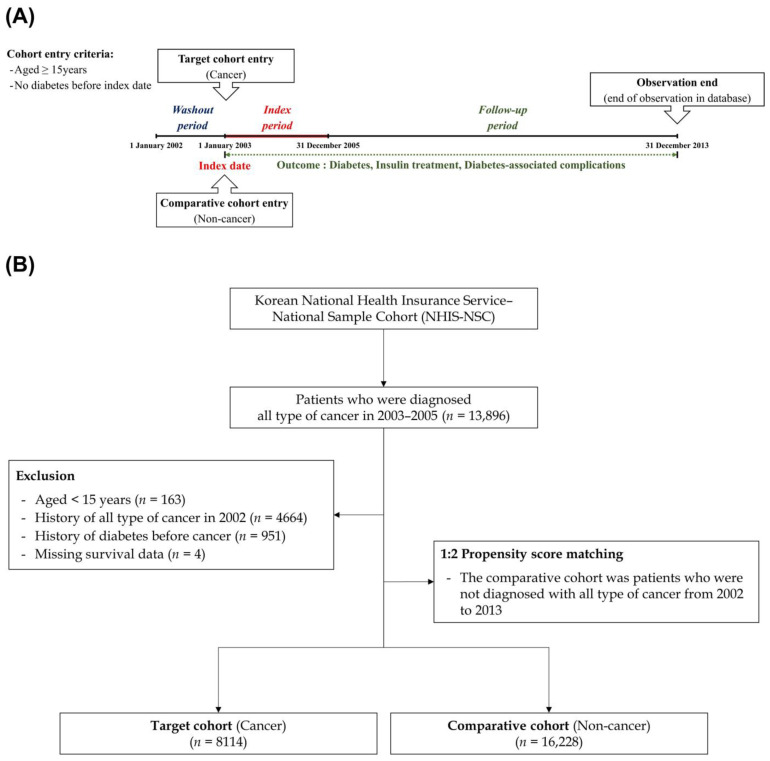
(**A**) Schematic description of the study design; (**B**) flow of enrollment of study participants.

**Figure 2 cancers-15-01094-f002:**
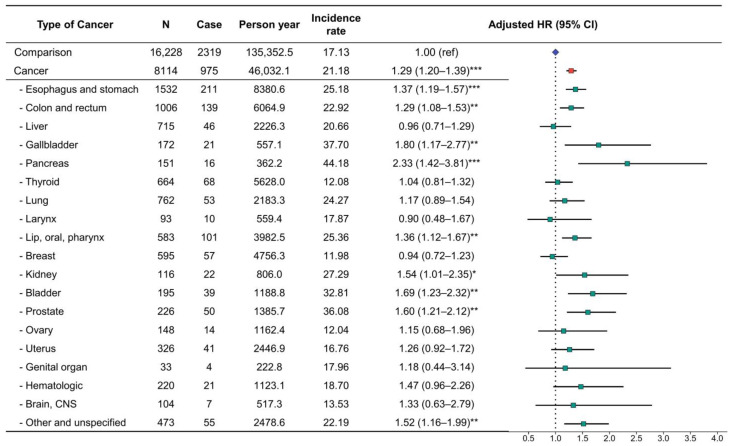
Incidence rate and hazard ratio of type 2 diabetes by cancer type. * *p* < 0.05, ** *p* < 0.010, and *** *p* < 0.001.

**Figure 3 cancers-15-01094-f003:**
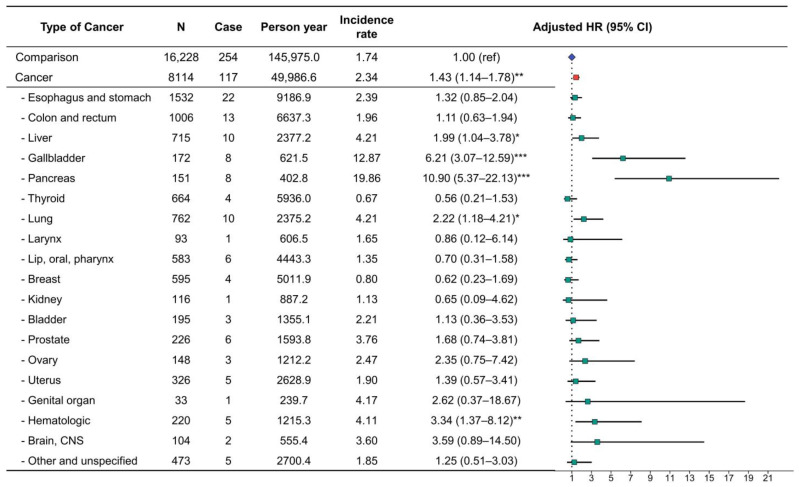
Incidence rate and adjusted hazard ratio for cases requiring insulin treatment in patients with type 2 diabetes by cancer type. * *p* < 0.05, ** *p* < 0.010, and *** *p* < 0.001.

**Figure 4 cancers-15-01094-f004:**
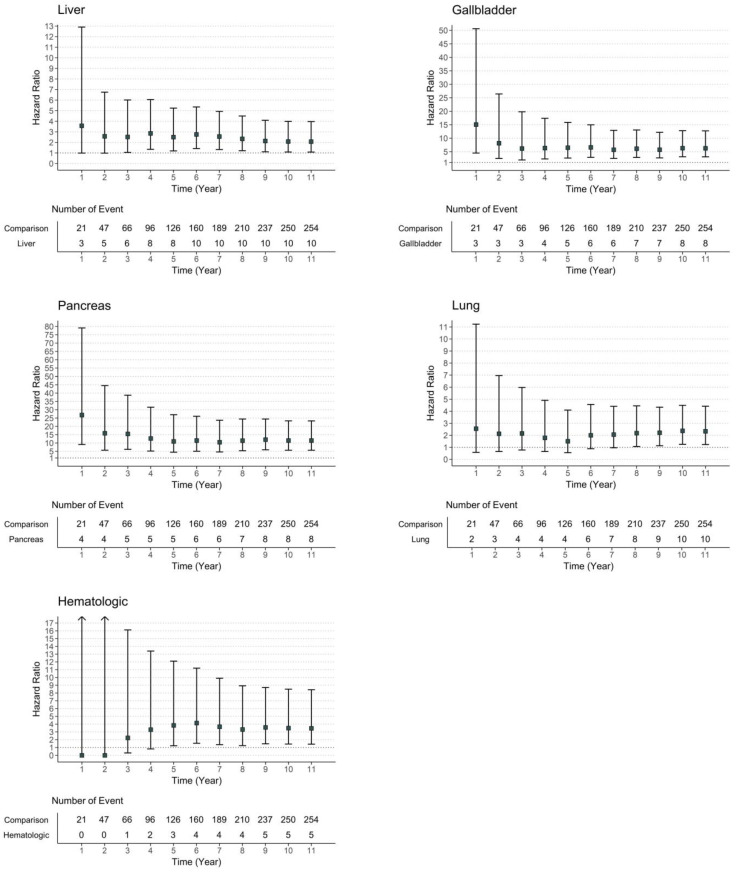
Risk of requiring insulin treatment in patients with liver, gallbladder, pancreas, lung, and hematologic malignancy by time.

**Figure 5 cancers-15-01094-f005:**
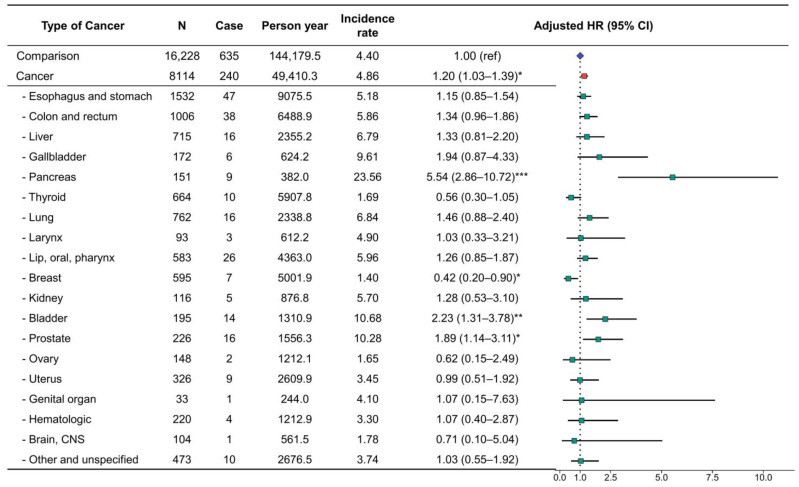
Incidence rate and adjusted hazard ratio for cases of diabetes-associated complications in patients with type 2 diabetes by cancer type. * *p* < 0.05, ** *p* < 0.010, and *** *p* < 0.001.

**Figure 6 cancers-15-01094-f006:**
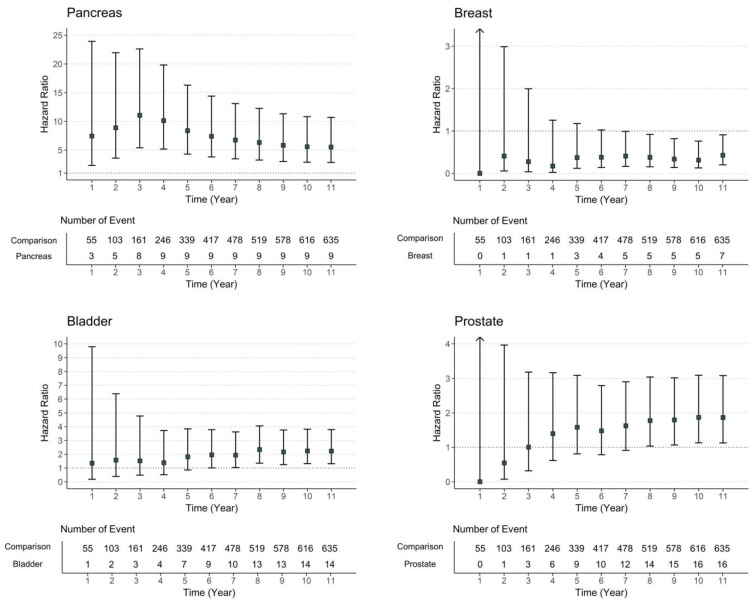
Risk of diabetes-associated complications in patients with pancreas, breast, bladder, and prostate malignancy by time.

**Figure 7 cancers-15-01094-f007:**
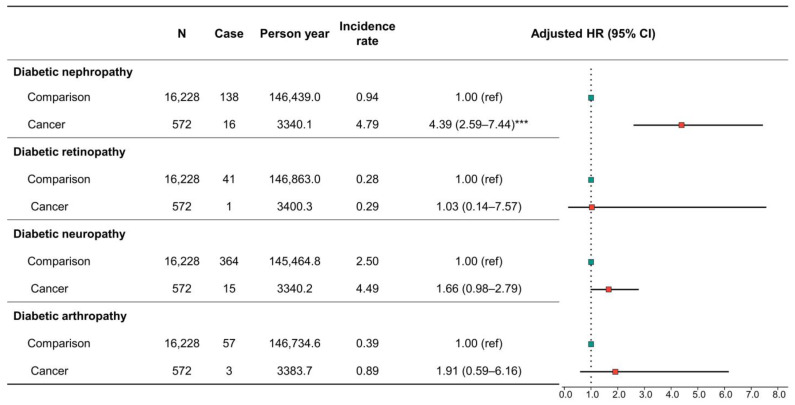
Subgroup analysis for incidence and risk of specific diabetes-associated complications in patients with specific cancer (pancreas, bladder, and prostate). *** *p* < 0.001.

**Table 1 cancers-15-01094-t001:** Sensitivity analysis: incidence and risk of incident diabetes associated with cancer patients who excluded those diagnosed with pancreatic cancer.

Variables	N	Case	Person Year	Incidence Rate	Unadjusted HR (95% CI)	Adjusted HR (95% CI)
Comparison	16,228	2319	135,352.5	17.13	1.00 (ref)	1.00 (ref)
Overall cancer	8114	975	46,032.1	21.18	1.21 (1.12–1.30) ***	1.29 (1.20–1.39) ***
Cancer without pancreatic cancer	7963	959	45,669.9	21.00	1.20 (1.11–1.29) ***	1.28 (1.19–1.38) ***

HR, hazard ratio; CI, confidence interval; *** *p* < 0.001.

**Table 2 cancers-15-01094-t002:** Sensitivity analysis: incidence and risk of insulin treatment in patients with cancer who excluded those diagnosed with pancreatic cancer.

Variables	N	Case	Person Year	Incidence Rate	Unadjusted HR (95% CI)	Adjusted HR (95% CI)
Comparison	16,228	254	145,975.0	1.74	1.00 (ref)	1.00 (ref)
Overall cancer	8114	117	49,986.6	2.34	1.33 (1.06–1.65) *	1.43 (1.14–1.78) **
Cancer without pancreatic cancer	7963	109	49,583.8	2.20	1.25 (1.00–1.56)	1.35 (1.07–1.69) **

HR, hazard ratio; CI, confidence interval; * *p* < 0.05 and ** *p* < 0.010.

## Data Availability

The original datasets are not publicly available because of KNHIS policies; however, if the corresponding author deems it reasonable, the approach of generating and/or analyzing data in the present study could be made available.

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
