# Peer review of "Analysis of the Incidence of Type 2 Diabetes, Requirement of Insulin Treatment, and Diabetes-Related Complications among Patients with Cancer"

_cancers, 2023, doi:10.3390/cancers15041094_

Round 1
Reviewer 1 Report
The manuscript of Lee et al. presents a retrospective nationwide population-based cohort study aimed at evaluating the risk of Type 2 diabetes among cancer patients. To achieve their goal, Authors analyzed data obtained from the national database of the Korean National Health Insurance Service. Authors employed log-rank test and Cox proportional hazards regression models to investigate whether cancer patients incurred enhanced risk rates to develop Type 2 diabetes. Authors followed up 8100 cases with diverse cancer diagnosis for a period of 8 years and evaluated the outcomes as 1) onset of diabetes, 2) application of insulin treatment and 3) the development of diabetes-associated complications. Authors compared these data to a control group of 16’200 individuals without cancer diagnosis. Based upon their findings, Authors conclude that individuals with cancer diagnosis show an increased incidence of Type 2 diabetes development and insulin requirements. Authors also report an enhanced risk to present diabetic nephropathy but no other complications in cancer patients. Authors support their manuscript by 7 Figures and 2 Tables and cites 43 references to put their findings in context.
The manuscript fits the scope of the “Cancers” and the manuscript is of interest for the readers of the journal. The manuscript is well written, the data are presented in a clear fashion. Authors address the limitations of their study in an honest manner but also point out the strong points that bring novel and significant contributions to this field.
This reviewer notes only the following issues:
1. A similar recent study in a Danish cohort also noted a highly elevated risk of diabetes in patients diagnosed with pancreatic cancer: Diabetes Care 2022;45:e105–e106 | https://doi.org/10.2337/dc22-0232. Authors should reference this article that would further confirm their findings.
2. Authors refer to a class of cancers as “GI track cancer”. Could Authors provide a more detailed analysis differentiating between “oesophageal-gastric” and “colorectal” cancers?
3. Minor issues:
- Page 2, Line 54: “pancreas” is misspelled
- Page 9: Figure 7 is marked as Figure 3.
Author Response
1. A similar recent study in a Danish cohort also noted a highly elevated risk of diabetes in patients diagnosed with pancreatic cancer: Diabetes Care 2022;45:e105–e106 | https://doi.org/10.2337/dc22-0232. Authors should reference this article that would further confirm their findings.
→ Response: First of all, we sincerely appreciate the evaluation of the referees. We agree with the reviewer’s advice. Thus, as suggested, we have cited the reference for the statement quoted.
2. Authors refer to a class of cancers as “GI track cancer”. Could Authors provide a more detailed analysis differentiating between “oesophageal-gastric” and “colorectal” cancers?
→ Response: We thank the reviewer for raising this point. We revised the GI tract as esophagogastric and colorectal cancer throughout the whole manuscript. However, all findings are still constant.
3. Minor issues:
Page 2, Line 54: “pancreas” is misspelled
Page 9: Figure 7 is marked as Figure 3.
→ Response: We modified these errors. Thank you so much.
Reviewer 2 Report
Thank you for the opportunity to review this article. This population-based study aimed to assess whether the risk of diabetes increased after the onset of cancer using a propensity-matching analysis. Precisely, this was a retrospective study and the authors used a dataset collected from the Korean National Health Insurance Service during the follow-up period (seems to be 10 years, but it is not clear). The subjects with pre-existing diabetes were excluded. In addition, the authors investigated the risk of insulin requirement and diabetes-associated complications in cancer survivors, compared with participants without cancer. The results suggest that subjects with cancer had increased incidence of type 2 diabetes and insulin requirement. In addition, some cancer types (pancreas, bladder, and prostate) were associated with diabetic nephropathy.
Generally, good work has been done, although the study has several limitations as it has been clearly highlighted by authors. The results are presented well. However, some parts are not clear and need to be improved.
Overall, the authors should be commended for writing an interesting article. The tables and figures are clear and useful to the readers. I recommend this article to be accepted after major revision.
Here are some suggestions that authors might find useful:
1. Duration of the follow-up period should be stated clearly in the abstract and through the manuscript;
2. In the abstract the statement “some cancer types (pancreas, bladder, and prostate) showed an increased risk of subsequent diabetic nephropathy” seems to be repetitive, please check;
3. I suggest the following sentences to be rewritten in order to make them more readable, it seems to be repetitive too: “Some studies have shown an increased incidence of type 2 diabetes in patients with specific cancers, including bladder, lung, and pancreas [7–9], whereas other studies revealed that specific cancers, such as breast, colorectal, and pancreas, are related to the risk of developing diabetes [10–12]. Additionally, some recent cohort studies described that patients diagnosed with cancer had a clear increase in the subsequent risk of diabetes [13,14].”
4. Please write clearly the following statement: “One systematic study reported that the decline in diabetes management seems to be primarily due to a shift in the primary concern of management from diabetes to cancer [19].”
5. In introduction section the authors stated: “However, only a few studies have been conducted on the risk of requiring insulin treatment and complications related to newly developed type 2 diabetes among patients with cancer based on a nationwide population.” Please include the appropriate references and consider to discuss briefly that today there are several approaches to treat diabetes (incretins, gliflozins, etc.; the insulins are the last therapeutic choice), stating that in the present retrospective study you are using data that have been recorded before the approval of the novel non-insulin antidiabetic drugs (or say something similar). Also, please discuss briefly in the discussion section if there is any similar study using novel antidiabetic drugs;
6. Please explain why was there a need for a written informed consent? This reviewer does not understand well your statement “because this study obtained secondary data from the KNHIS database after de-identification.” Was there any biological data?
7. Regarding the study design please consider the following:
a) Please check the schematic picture. Is it correct that the follow-up period stared in the middle of index period (Jun 2003)?
b) Why did you decide 15 years old to be an age limit to include/exclude participants? If consents were obtained, does it mean that parental consents were required too?
c) Why propensity score matching is 1:2? Then, the authors mentioned: “We randomly selected propensity score-matched participants from the remaining cohort registered in the database as two participants without cancer for each patient with cancer. Thus, each participant in the comparison group was matched with each patient in the cancer group based on all independent variables and year of enrollment (cancer diagnosis).” Each participant in the comparison group was matched with each patient in the cancer group? Is not this contradictory to the score matching 1:2? Further, please specify which independent variables. Also, please specify in details the years of enrollment (cancer diagnosis, including the stage of cancer if possible) as well as duration of the follow-up period;
d) Are you sure that using the term “a washout period” is correct in the case of this study?
e) Please rewrite and explain better what do you think by saying: “During this process, we carefully reviewed the C code because it has implications for additional benefits for patients with respect to insurance coverage.” In addition, it is not clear if the cancer stadium has been taken in consideration? Please explain it and add in the text;
f) The authors stated several primary endpoints. There is usually only one primary and several secondary endpoints. Please explain it and consider to correct accordingly;
g) Similarly, you stated that all-cause mortality was investigated too, but you did not report any data. Please discuss;
h) What do you mean by saying: “For patients who had no events”? Which events?
8. In the paragraph Independent Variables, the authors mentioned, among others, residence, and household income, as well as comorbidities. Please explain why did you consider residence, and household income and how it could influence your findings? Also, please discuss in the discussion section about comorbidities (which comorbidities have been found) and how they may influence the obtained results? Do you think diabetic comorbidities?
9. I suggest to reword “appeared over a certain period”. What does a certain period mean? The follow up period? Which specific disease or “other incident event”? Please be specific with your statements;
10. Similarly, please define “specific disease-free periods”;
11. The authors stated: “To identify the risk of insulin requirement, we selected patients with diabetes who were receiving insulin treatment.” What was previous therapeutic approach (before insulin)? Lifestyle changes? Metformin?
12. Then, the authors stated: “Thus, hyperglycemia or poor blood sugar control may contribute to the need for insulin treatment in cancer survivors”. What can be suggested to clinicians in order to have glycemia level under control and to avoid use of insulin? Could diet/nutraceuticals be combined with chemotherapy and decrease the adverse effects, including better glycemic control? What about novel anti-diabetes agents? Please discuss;
13. The following sentences are somewhat confusing, please try to reword this part making it more readable and understandable: “Third, the two groups were compared by limiting the severity of diabetes to whether insulin was used. Since insulin is often used for patients whose blood sugar is not controlled, it is difficult to directly determine the degree of blood sugar control according to the severity of diabetes.”
14. The authors stated clearly both the strengths and limitations of the present study. However, please discuss briefly may lifestyle behavior differ between different regions? Was there any data about presence of obesity or dyslipidemia or other factors that also could be associated with the onset of diabetes?
15. Conclusion should be specific highlighting the novelty of the present study. For instance, you get that some cancer types are significantly associated with an increased risk of diabetic nephropathy, but after that you state that cancer survivors are at high risk for developing diabetic nephropathy. Please correct;
Some references (for example n. 16) are too old. If it is not extremity necessary to cite the work from 1995, the reference should be changed appropriately.
Author Response
- Duration of the follow-up period should be stated clearly in the abstract and through the manuscript.
Response: We agree with the reviewer’s advice. We have therefore stated the follow-up period clearly in the abstract and throughout the manuscript.
- In the abstract the statement “some cancer types (pancreas, bladder, and prostate) showed an increased risk of subsequent diabetic nephropathy” seems to be repetitive, please check.
Response: We apologize for the confusion. This statement refers to a different finding. To improve the clarity of the text, we have slightly modified this sentence.
“Although there was no significant association between diabetes-associated complications and overall cancer diagnosis, specific cancer types (pancreas, bladder, and prostate) showed an increased risk of subsequent diabetic nephropathy.” (page 1, line 33-35)
- I suggest the following sentences to be rewritten in order to make them more readable, it seems to be repetitive too: “Some studies have shown an increased incidence of type 2 diabetes in patients with specific cancers, including bladder, lung, and pancreas [7–9], whereas other studies revealed that specific cancers, such as breast, colorectal, and pancreas, are related to the risk of developing diabetes [10–12]. Additionally, some recent cohort studies described that patients diagnosed with cancer had a clear increase in the subsequent risk of diabetes [13,14].”
Response: We thank the reviewer for the advice. As suggested, we have modified these sentences as follows: “Although some studies have reported that specific cancers, such as breast, colorectal, and pancreas, are related to the risk of developing diabetes [7-9], some cross-sectional studies [10,11] and cohort studies [12-14] described that patients diagnosed with cancer had a clear increase in the subsequent risk of type 2 diabetes.” (page 2, line 57-60)
- Please write clearly the following statement: “One systematic study reported that the decline in diabetes management seems to be primarily due to a shift in the primary concern of management from diabetes to cancer [19].”
Response: We apologize for the lack of clarity in this statement. To improve the readability of this statement, we have rephrased this sentence as follows: “Previously, a systematic study reported a general decline in diabetes management after a cancer diagnosis because the primary concern shifted from diabetes to cancer.” (page 2, line 63-65)
- In introduction section the authors stated: “However, only a few studies have been conducted on the risk of requiring insulin treatment and complications related to newly developed type 2 diabetes among patients with cancer based on a nationwide population.” Please include the appropriate references and consider to discuss briefly that today there are several approaches to treat diabetes (incretins, gliflozins, etc.; the insulins are the last therapeutic choice), stating that in the present retrospective study you are using data that have been recorded before the approval of the novel non-insulin antidiabetic drugs (or say something similar). Also, please discuss briefly in the discussion section if there is any similar study using novel antidiabetic drugs;
Response: As suggested, we have cited the reference for the statement quoted. Additionally, we have added further description regarding the current methods used to treat diabetes and the use of insulin as a last therapeutic choice in the Introduction section: “Current consensus recommendations for the management of hyperglycemia in patients with type 2 diabetes suggest lifestyle changes and monotherapy, preferably with metformin. However, if the glycated hemoglobin (HbA1c) target is still not achieved, combination injectable therapy including basal insulin can be considered to obtain glycemic control.” (Page 2, line 52-56)
However, the database used consisted of health records between 2002-2013, before the approval of the novel non-insulin antidiabetic drugs; thus, we could not access novel non-insulin antidiabetic drugs.
- Please explain why was there a need for a written informed consent? This reviewer does not understand well your statement “because this study obtained secondary data from the KNHIS database after de-identification.” Was there any biological data
Response: To clarify, we stated “The IRB waived the need for written informed consent", which means that this study does not need written informed consent.
- Regarding the study design please consider the following: a) Please check the schematic picture. Is it correct that the follow-up period stared in the middle of index period (Jun 2003)?
Response: We revised the schematic picture and accordingly modified Figure 1A.
b) Why did you decide 15 years old to be an age limit to include/exclude participants? If consents were obtained, does it mean that parental consents were required too?
Response: We thank the reviewer for raising this point. This database (KNHIS) provides age groupings at 5-year intervals for de-identification. Therefore, we arbitrarily determined 15 years of age as the earliest age at which the onset of type 2 diabetes may occur after the cancer diagnosis.
c) Why propensity score matching is 1:2? Then, the authors mentioned: “We randomly selected propensity score-matched participants from the remaining cohort registered in the database as two participants without cancer for each patient with cancer. Thus, each participant in the comparison group was matched with each patient in the cancer group based on all independent variables and year of enrollment (cancer diagnosis).” Each participant in the comparison group was matched with each patient in the cancer group? Is not this contradictory to the score matching 1:2? Further, please specify which independent variables. Also, please specify in details the years of enrollment (cancer diagnosis, including the stage of cancer if possible) as well as duration of the follow-up period;
Response: Propensity score matching is a method of matching the distribution of the dependent variable selected between two groups to be compared in a study design where random selection cannot be applied and is a statistical analysis method to reduce selection bias. We observed that there were no significant differences in all the independent variables (age, sex, residence, household income, and comorbidities) between the two groups, indicating that each variable was appropriately matched. We also confirmed the matching status using the balance plot. Additionally, through propensity score matching, 1:n matching can be used to increase precision in cohort studies. A previous study recommended a variable ratio, parallel, balanced 1:n, nearest neighbor approach that increases precision over 1:1 matching at a small cost in bias (Pharmacoepidemiol Drug Saf. 2012 May;21 Suppl 2:69-80. Doi: 10.1002/pds.3263). Also, 1:2 or 1:4 PSM with nearest neighbor is known to be most effective in the claim data study.
d) Are you sure that using the term “a washout period” is correct in the case of this study?
Response: In the claim data study, the "washout period" was achieved by identifying and excluding patients that exhibited previous events in claims data during a pre-index date. Thus, we believe the term "washout period” is used correctly in this study.
e) Please rewrite and explain better what do you think by saying: “During this process, we carefully reviewed the C code because it has implications for additional benefits for patients with respect to insurance coverage.” In addition, it is not clear if the cancer stadium has been taken in consideration? Please explain it and add in the text;
Response: We apologize for the unclear language. We wanted to convey that in actual clinical practice, there are cases in which C-code is entered as a diagnosis code before biopsy results for insurance coverage, which can lead to confusion in determining the diagnosis date. Therefore, we conducted meticulous reviews to set the diagnosis date at the time of histological diagnosis. We have added more description about this issue as follows: “During this process, we carefully reviewed the C code because sometimes it is entered as a diagnosis code before biopsy results for insurance coverage, which can lead to confusion in determining the diagnosis date.” (page 4, line 108-111)
f) The authors stated several primary endpoints. There is usually only one primary and several secondary endpoints. Please explain it and consider to correct accordingly;
Response: We have clarified this issue in the revised version as follows:
“In this study, we defined the primary endpoints as incident development of specific events, such as type 2 diabetes, initial insulin treatment, and diabetes-associated complications. Additionally, we performed follow until the secondary endpoint, all-cause mortality, was reached. For patients who had no specific events and were alive on December 31, 2013 (the final follow-up day for this database), the data were censored at this time point (Supplementary Tables 1 and 2).” (page 4, line 119-124)
g) Similarly, you stated that all-cause mortality was investigated too, but you did not report any data. Please discuss;
Response: We apologize for the confusion and the unclear text. Our intended meaning is that we censored patients’ data at this time point for patients who were diagnosed with any primary end point or died of any cause. We have clarified the text as follows: “For patients who had no specific events and were alive on December 31, 2013 (the final follow-up day for this database), the data were censored at this time point (Supplementary Tables 1 and 2).” (page 4, line 122-124)
h) What do you mean by saying: “For patients who had no events”? Which events?
Response: We apologize for the unclear text. We have modified the sentence to improve clarity.
- In the paragraph Independent Variables, the authors mentioned, among others, residence, and household income, as well as comorbidities. Please explain why did you consider residence, and household income and how it could influence your findings? Also, please discuss in the discussion section about comorbidities (which comorbidities have been found) and how they may influence the obtained results? Do you think diabetic comorbidities?
Response: We thank the reviewer for raising this point. In this study, we wanted to minimize the effect of lifestyle and behavior differences. So, we matched the income status and living regions between the two groups because these variables are thought be one of the determining factors for differences in lifestyle and behavior.
Additionally, our main goal was to investigate the risk of incident development for type 2 diabetes, requirement of insulin, and diabetes-associated complication events. Thus, we wanted to remove the effect of all diseases other than cancer. Therefore, we matched the overall comorbidity status between the two groups using the Charlson comorbidity index (CCI). The CCI, the most ubiquitous comorbid risk score, predicts one-year mortality among hospitalized patients and provides a single aggregate measure of patient comorbidity. In the claim data study, most investigators have used the CCI to match the comorbidities status between control and experimental groups (Psychother Psychosom 2022;91:8–35, Pharmacoepidemiol Drug Saf. 2021 May;30(5):582-593. BMC Med Res Methodol. 2019;19(1):115). To improve the clarity of the text, we have added further explanation about this issue in the Discussion section.
“In addition, we matched several important variables that could affect the incidence of diabetes in both groups using propensity scores. Thus, we could minimize the effect of lifestyle and behavior differences between different age groups, sexes, income groups, and regions. Additionally, we wanted to eliminate the effect of diseases other than cancer on the analysis of the risk of incident development for type 2 diabetes, requirement of insulin, and diabetes-associated complication events. Thus, in this study, we matched the overall comorbidity status between the two groups using the CCI. The CCI, the most ubiquitous comorbidity risk score, predicts one-year mortality among hospitalized patients and provides a single aggregate measure of patient comorbidity. Several recent claim data studies have also used the CCI to match the comorbidities status between the control and experimental groups [44-48].” (page 12-13, line 337-347)
- I suggest to reword “appeared over a certain period”. What does a certain period mean? The follow up period? Which specific disease or “other incident event”? Please be specific with your statements;
Response: We have modified this sentence to improve clarity as follows: “We assessed the incidence rate as a measure of the frequency at which type 2 diabetes or other incident events (initial insulin treatment and diagnosis of diabetes-associated complications) appeared over the follow-up period.” (page 4, line 137-139)
- Similarly, please define “specific disease-free periods”;
Response: We removed the Kaplan–Meier survival curve during the manuscript writing process. Therefore, the phrase is no longer needed in the methodology and was removed. Thanks for pointing out our mistake.
- The authors stated: “To identify the risk of insulin requirement, we selected patients with diabetes who were receiving insulin treatment.” What was previous therapeutic approach (before insulin)? Lifestyle changes? Metformin?
Response: According to the recommendations of the American Diabetes Association, insulin treatment should be given if HbA1c does not reach the appropriate target despite the use of two or more oral hypoglycemic agents. Thus, we assumed that patients who need insulin may suffer from a more severe form of type 2 diabetes. That is why we focused on the risk ratio of insulin treatment in patients with cancer.
- Then, the authors stated: “Thus, hyperglycemia or poor blood sugar control may contribute to the need for insulin treatment in cancer survivors”. What can be suggested to clinicians in order to have glycemia level under control and to avoid use of insulin? Could diet/nutraceuticals be combined with chemotherapy and decrease the adverse effects, including better glycemic control? What about novel anti-diabetes agents? Please discuss;
Response: Thank you for your nice comments. In general, if blood sugar is not controlled in diabetic patients, additional insulin is used to control blood sugar. Specifically, in cancer survivors, cancer-related treatments mentioned earlier in the acute phase of treatment (such as anticancer drugs and steroids) can increase the risk of developing diabetes or poor blood sugar control. So, the sentence “Thus, hyperglycemia or poor blood sugar~” was written to explain this. In order to reduce the confusion of the sentence, we modified the sentence as follows: “Thus, hyperglycemia or poor blood sugar control for cancer-specific treatment may contribute to the need for insulin treatment in cancer survivors.”
- The following sentences are somewhat confusing, please try to reword this part making it more readable and understandable: “Third, the two groups were compared by limiting the severity of diabetes to whether insulin was used. Since insulin is often used for patients whose blood sugar is not controlled, it is difficult to directly determine the degree of blood sugar control according to the severity of diabetes.”
Response: We have modified this sentence to make it clearer:
“Third, the two groups were compared by determining the severity of diabetes only based on whether insulin was used. According to the recommendations of the American Diabetes Association, insulin treatment should be given if HbA1c does not reach the appropriate target despite the use of two or more oral hypoglycemic agents. However, in this study, we could not directly determine the degree of blood sugar control level because of the absence of HbA1c levels in our database.” (page 12, line 325-330)
- The authors stated clearly both the strengths and limitations of the present study. However, please discuss briefly may lifestyle behavior differ between different regions? Was there any data about presence of obesity or dyslipidemia or other factors that also could be associated with the onset of diabetes?
Response: We completely agree with your advice. Thus, to minimize the effect of lifestyle behavior between different regions, we selected age, sex, income level, and residence area as the independent variables and matched between the two groups. Additionally, due to the limitation of claims data, we did not have access to personal health data on smoking, alcohol consumption, BMI, and any lab data. We have addressed these issues in the Discussion.
“Thus, we could minimize the effect of lifestyle and behavior differences between different age groups, sexes, income groups, and regions. Additionally, we wanted to eliminate the effect of diseases other than cancer on the analysis of the risk of incident development for type 2 diabetes, requirement of insulin, and diabetes-associated complication events.” (page 13, line 338-342)
- Conclusion should be specific highlighting the novelty of the present study. For instance, you get that some cancer types are significantly associated with an increased risk of diabetic nephropathy, but after that you state that cancer survivors are at high risk for developing diabetic nephropathy. Please correct;
àResponse: Thanks for your comments. We have modified the Conclusion as follows: “Therefore, we suggest that clinicians should consider that patients with cancer may have a higher risk of developing type 2 diabetes and require insulin treatment. Additionally, clinicians should be aware of the potential risk of developing diabetic nephropathy in patients who suffer from pancreatic, bladder, and prostate cancer.” (page 13, line 356-360)
Some references (for example n. 16) are too old. If it is not extremity necessary to cite the work from 1995, the reference should be changed appropriately.
Response: Thank you for noting this. We have changed this reference accordingly.

Round 2
Reviewer 2 Report
The authors have carefully addressed all my comments and I am satisfied with the improvements.
However, I have few minor comments:
1. Please include the references after the statement: “Current consensus recommendations for the management of hyperglycemia in patients with type 2 diabetes suggest lifestyle changes and monotherapy, preferably with metformin. However, if the glycated hemoglobin (HbA1c) target is still not achieved, combination injectable therapy including basal insulin can be considered to obtain glycemic control.”
2. I suggest the following to be mentioned in the material and methods section: “The database used the records between 2002-2013, before the approval and use of the novel non-insulin antidiabetic drugs.”
3. In addition, I thing that it could be nice to add a brief comment in the discussion section if there is any similar study using novel antidiabetic drugs and if something could be changed using novel approaches in order to improve control of glycemic level in cancer patients.
4. Finally, it remained not clear if the cancer stadium has been taken in consideration. Thank you.
Author Response
- Please include the references after the statement: “Current consensus recommendations for the management of hyperglycemia in patients with type 2 diabetes suggest lifestyle changes and monotherapy, preferably with metformin. However, if the glycated hemoglobin (HbA1c) target is still not achieved, combination injectable therapy including basal insulin can be considered to obtain glycemic control.”
Response: As suggested, we have cited the reference for the statement quoted.
- I suggest the following to be mentioned in the material and methods section: “The database used the records between 2002-2013, before the approval and use of the novel non-insulin antidiabetic drugs.”
Response: We thank the reviewer for the advice. As suggested, we added this.
- In addition, I thing that it could be nice to add a brief comment in the discussion section if there is any similar study using novel antidiabetic drugs and if something could be changed using novel approaches in order to improve control of glycemic level in cancer patients.
Response: We agree with the reviewer’s advice. However, we could not find any similar study using novel antidiabetic drugs. It probably is because the time after the release of novel antidiabetic drugs does not have enough period to investigate a long-term cohort study.
- Finally, it remained not clear if the cancer stadium has been taken in consideration. Thank you.
Response: We apologize for the lack of clarity in this statement. Our cohort database only includes the diagnostic code but it does not provide cancer characteristics such as histologic type, stage at diagnosis, or classification according to the TNM classification system. Thus, we added this issue in the section of Discussion as a limitation.